# Pillararene-Based Supramolecular Polymers for Cancer Therapy

**DOI:** 10.3390/molecules28031470

**Published:** 2023-02-03

**Authors:** Miaomiao Yan, Jiong Zhou

**Affiliations:** Department of Chemistry, College of Sciences, Northeastern University, Shenyang 110819, China

**Keywords:** pillararene, supramolecular polymers, cancer therapy, supramolecular chemistry

## Abstract

Supramolecular polymers have attracted considerable interest due to their intriguing features and functions. The dynamic reversibility of noncovalent interactions endows supramolecular polymers with tunable physicochemical properties, self-healing, and externally stimulated responses. Among them, pillararene-based supramolecular polymers show great potential for biomedical applications due to their fascinating host–guest interactions and easy modification. Herein, we summarize the state of the art of pillararene-based supramolecular polymers for cancer therapy and illustrate its developmental trend and future perspective.

## 1. Introduction

Cancer has become one of the “biggest killers” of human beings worldwide [1,2,3,4,5]. Cancer poses a threat to human health, claiming millions of lives each year [6,7,8,9,10]. Up to date, due to the malignant proliferation and complexity of tumors, existing cancer therapies generally destroy cancer cells, accompanied by some side effects [11,12]. Radiotherapy, chemotherapy, and surgical excision are generally considered the main treatments for cancer [13,14]. However, radiotherapy is a kind of local therapy that will produce a certain radiation reaction, which often brings certain toxic side effects and irreversible local damage [15]. In contrast, surgical resection may be a more direct approach to treat cancer. However, it causes huge distress to cancer patients and risks tumor cell migration [16]. As a systemic treatment, chemotherapy can kill local or distant metastatic tumors, but serious side effects on the body are inevitable [17,18]. Due to the limitations and side effects of these three traditional therapies, exploring new strategies for cancer therapy is urgently needed [19,20,21,22,23,24,25].

In recent years, the application of supramolecular chemistry in the field of biomedicine has attracted wide attention [26,27,28,29,30,31,32,33,34]. Supramolecular chemistry is ”chemistry beyond the molecule” [35,36,37,38,39,40]. Among them, supramolecular polymers not only combine the merits of supramolecular chemistry and polymer materials, but also possess novel structures and functions [41,42,43,44]. Supramolecular polymers refer to the ones based on monomeric units held together by directional and reversible noncovalent interactions [45,46]. Usually, supramolecular polymers are obtained via the self-assembly of building units, such as the noncovalent polymerization of covalent monomers and noncovalent crosslinking of linear polymers [47,48]. Noncovalent interactions, such as hydrogen bonding, host–guest interactions, and metal-ligand coordination, have been exploited to drive the self-assembly of building blocks [49,50,51]. The dynamic reversibility of noncovalent interactions endows supramolecular polymers with excellent properties, such as external stimulus response, self-healing, and degradability [52,53,54,55,56]. Compared with traditional polymers, it is much easier for supramolecular polymers to adjust their chemical and physical properties by changing solvents, concentrations, temperatures, or introducing new stimuli-responsive groups [57,58].

Pillararenes are the fifth generation of macrocyclic hosts next to crown ethers, cyclodextrins, calixarenes, and cucurbiturils [59,60,61,62,63,64,65,66,67,68,69,70,71], which are connected by methylene bridges at *para*-positions with hydroquinone as the repeating unit (Figure 1) [72,73,74]. The electron-rich cavity and unique pillar-shaped structure of pillararenes endow them with an excellent ability to bind guests [75,76,77]. In addition, various functional groups can be easily attached to the rim of pillararenes for their facile preparation and flexible modification [78,79]. The fascinating host–guest interaction and easy modification of pillararenes make them ideal materials for the preparation of supramolecular polymers [80,81,82,83]. Pillararene-based supramolecular polymers play an important role in cancer therapy, which can effectively solve some limitations of traditional materials in clinical applications [84,85]. The structure of supramolecular polymers may be controlled by the high self-selectivity based on host–guest interactions, and the rich environmental responsiveness based on noncovalent interactions can trigger the release of anticancer drugs through various stimuli [86,87].

In this review, we summarize the research progress of pillararene-based supramolecular polymers for cancer therapy over the past few years, including linear supramolecular polymers, branched supramolecular polymers, crosslinked supramolecular polymers, and supramolecular block copolymers. Moreover, the development prospects and challenges of pillararene-based supramolecular polymers for cancer therapy are extensively discussed. It is expected that this work will provide a reference for researchers interested in fields such as pillararene or cancer therapy.

## 2. Pillararene-Based Linear Supramolecular Polymers for Cancer Therapy

Linear supramolecular polymers, known as main-chain supramolecular polymers, are composed of monomers with bifunctional groups linked by noncovalent bonds [88,89]. Linear supramolecular polymers are the most common supramolecular polymers, which have several types such as AA, AB, AA/BB, and ABBA types [90,91]. The dynamic and reversible noncovalent interactions of pillararene endow the supramolecular polymers with outstanding stimuli-responsive features [92]. In recent years, pillararene-based functional supramolecular polymers as stimuli-responsive materials have been widely used in the fields of biology, medicine, and materials science [93,94]. Especially, pillararene-based linear supramolecular polymers have been recognized as effective materials for cancer therapy [95,96,97,98].

Photodynamic therapy (PDT) has become a cancer therapy strategy in the spotlight due to its advantages of low side effects, high selectivity, and low drug resistance [99,100,101,102]. It uses photosensitizers (PSs) and molecular oxygen to generate cytotoxic reactive oxygen species (ROS) to kill tumor cells under the irradiation of specific wavelengths. Compared with type-II PSs that generate singlet oxygen through energy transfer, type-I PSs are more advantageous in PDT because they can reduce the dependence of tumors on O_2_, and eliminate tumor cells efficiently even under hypoxic conditions [103,104,105]. Yang and co-workers prepared supramolecular polymeric PSs using a classical type-II PS (iodide BODIPY) as the guest molecule and electron-rich bispillar[5]arene (BP5A) as the macrocyclic host and electron donor [106]. The host–guest interaction shortened the distance between the BODIPY and the BP5A, which promoted the electron transfer of the BP5A to the BODIPY, resulting in the generation of a superoxide radical (O2^−∙^) through the type-I mechanism (Figure 2I). Antitumor studies showed that these supramolecular PSs had almost no dark toxicity under normoxic and hypoxic environments. Supramolecular PSs could kill HeLa cells effectively after irradiating with light for 10 min even under a hypoxic environment (Figure 2II), and exhibited an outstanding tumor-selective fluorescence imaging effect (Figure 2III). This work utilized host–guest interactions to convert type-II PSs into type-I PSs, achieving the effective regulation of both type-I and type-II mechanisms.

Monotherapy does not meet the clinical need for the effective treatment of cancer [107,108]. Thus, the synergistic combination strategy of multiple therapeutic modalities has become one of the main means to treat tumors [109,110]. Wang, Yao, and co-workers developed a pillar[5]arene-based supramolecular therapeutic nanoplatform (SP/GOx NPs) for synergistic chemo–chemodynamic therapy (Figure 3I) [111]. SP/GOx NPs could be easily loaded with DOX and modified target molecules (FA-Py) on their surfaces due to excellent host–guest properties. As the generated FA-Py/SP/GOx/Dox NPs entered the blood circulation, FA-Py could effectively target cancer cells and GOx could catalyze the overexpression of glucose in cancer cells to produce H_2_O_2_. H_2_O_2_ would form ∙OH under the catalysis of bridged ferrocene units. Thereby, the cancer cells were successfully killed and chemodynamic therapy was realized. In addition, the loaded DOX molecules were released in the acidic microenvironment to achieve synergistic chemotherapy. There was no significant change in the body weight of the mice after different treatments, showing the excellent biocompatibility and safety of pillar[5]arene-based nanomaterials (Figure 3II). Moreover, FA-Py/SP/GOx/Dox NPs showed the best antitumor effect, which illustrated the important role of the target molecule, and also demonstrated the outstanding therapeutic effect of targeted/synergistic chemo–chemodynamic therapy (Figure 3III). This study demonstrated that pillar[5]arene-based supramolecular polymers could be used as effective materials for targeted and combined chemo–chemodynamic therapy.

## 3. Pillararene-Based Branched Supramolecular Polymers for Cancer Therapy

In addition to the pillararene-based linear supramolecular polymers, pillararene-based branched supramolecular polymers have been employed for tumor diagnosis and therapy [112,113]. Branched supramolecular polymers have multiple branched sites, which can be constructed through noncovalent interactions or covalent synthesis [114,115]. The structure of branched supramolecular polymers can be subdivided into star, brush, hyperbranched, and other structures [116]. Pillararene-based branched supramolecular polymers with abundant terminal groups and reversible and tunable properties have potential applications in biomedicine and materials chemistry, especially in cancer therapy.

Brush polymers are a class of branched or grafted polymers, whose polymeric side-chains are usually attached to a linear backbone [117]. The distinctive topological structure and relatively low critical aggregation concentration endow brush polymers with incomparable advantages in biomedical applications [118,119]. In addition, due to the compact structure, a large number of drugs can be loaded into the self-assembly formed by brush polymers, thus providing sufficient drug concentration at the active site [120]. There are many brush polymers used in cancer therapy to improve anticancer efficacy and reduce side effects [121]. For example, Chen, Huang, and co-workers also synthesized another brush supramolecular polymer based on pillar[5]arene for targeted drug delivery (Figure 4a(I)) [122]. DOX, a widely used chemotherapeutic drug, was encapsulated into the interior of supramolecular nanoparticles constructed by a pillar[5]arene-based amphiphilic supramolecular brush copolymer. Under the action of intracellular reductase and low pH environment, the loaded DOX was released. The energy transfer relay effect between DOX and tetraphenylethene was interrupted, enabling the in situ visualization of the drug release by observing the location and magnitude of the energy transfer-dependent fluorescence changes. Additionally, supramolecular nanoparticles reduced the aggregation of nanocarriers through the formation of a “brush-like” structure, which could increase the blood circulation time. Thus, the drug was more likely to reach the target location before it was recognized and internalized by phagocytes and excreted by the reticuloendothelial system. The blood circulation time of DOX-loaded nanoparticles was much longer than that of free DOX, thus giving nanoparticles more chances to extravasate from tumor vessels (Figure 4a(II)). In vivo experiments indicated that DOX-loaded nanoparticles showed an excellent anticancer activity with negligible systemic toxicity (Figure 4b(III)).

Stimuli-responsive polymers have attracted considerable attention for their potential applications in biomedical materials [123,124,125,126]. Among various stimuli, light is a particularly attractive option due to its characteristics of convenience, cleanliness, and controllability [127,128,129]. Owing to its specific spatial and temporal controllability, photomedicine has been widely used in cancer therapy [130,131]. Tong, Jiang, and co-workers prepared a host–guest complex between pillar[6]arene and azobenzene-containing block copolymers (Figure 4b(I)) [132]. The resulting supramolecular complex was further self-assembled to form vesicles in PBS solution, where the release rate of DOX·HCl could be regulated by pH and light stimulation. Because of the good biocompatibility of vesicles, MTT assay proved that the cell viability was close to 100% at concentrations ranging from 100 to 500 μg m L^−1^, indicating that the nanocarriers had low cytotoxicity (Figure 4b(II)). Compared with free DOX·HCl, DOX·HCl-loaded vesicles showed lower cytotoxicity, which meant that encapsulating DOX·HCl in vesicles could reduce the toxicity of DOX·HCl (Figure 4b(III)). This pillararene-based drug delivery system with dual stimuli-responsiveness provided a new strategy for cancer therapy.

**Figure 4 molecules-28-01470-f004:**
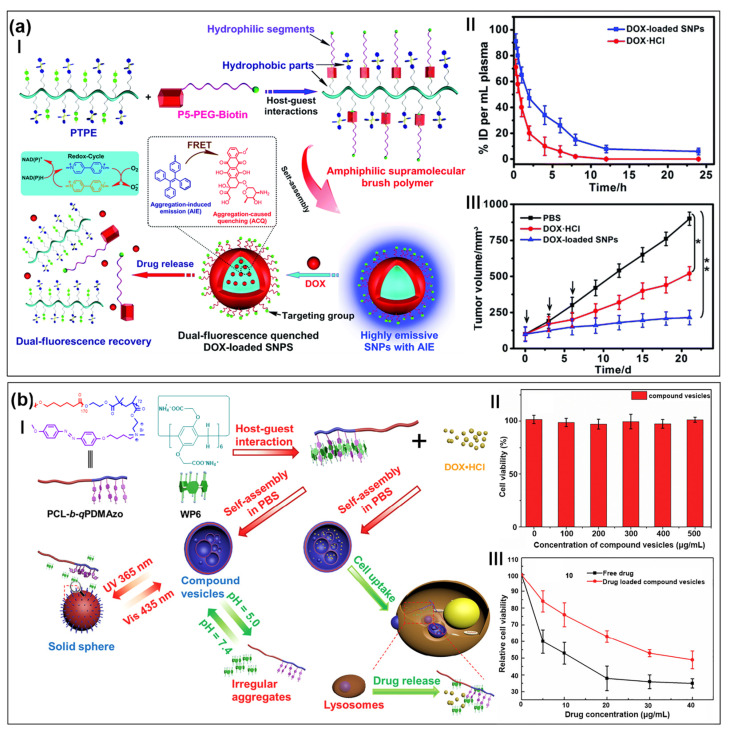
(**a**) Schematic illustration of pillar[5]arene-based amphiphilic supramolecular brush polymers for targeted drug delivery (**I**). Blood circulation time of mice after injection of DOX·HCl and DOX-loaded nanoparticles (**II**). Tumor growth inhibition curves in tumor-bearing mice with different formulations (**III**) (* *p* < 0.05, ** *p* < 0.01). (reproduced with permission of Royal Society of Chemistry from ref. [122]). (**b**) Schematic illustration of dual-responsive controlled assembly and release of drugs in vesicles (**I**). Cytotoxicity of vesicles at different concentrations to MCF-7 cells (**II**). Effects of DOX·HCl-loaded vesicles and free drugs on proliferation of MCF-7 cells in vitro (**III**). (reproduced with permission of Elsevier from ref. [132]).

The abnormal proliferation of the pathogenic bacteria Fusobacterium nucleatum (*F. nucleatum*) around colorectal cancer is usually an important cause of chemotherapy failure and drug resistance [133]. Multifunctional cationic quaternary ammonium materials are widely used in antibacterial and anticancer fields [134,135]. Chen, Yang, Gao, and co-workers constructed a dendritic supramolecular nanoparticle based on quaternary ammonium-polyamidoamine–azobenzene (Q-P-A) and carboxylatopillar[5]arene (CP[5]A) for antibacterial and antitumor therapies, namely Q-P-A@CP[5]A (Figure 5I) [136]. The -N^+^CH_3_ group on the surface of Q-P-A was accommodated in the cavity of CP[5]A, which greatly improved the biocompatibility of Q-P-A@CP[5]A. Under a pathological situation, CP[5]A could be separated from the -N^+^CH_3_ group, allowing for effective antibacterial and anticancer therapies. Compared with other treatment groups, the Q-P-A@CP[5]A and *F. nucleatum* treatment group showed enhanced therapeutic effects, suggesting that -N^+^CH_3_ in Q-P-A@CP[5]A could effectively limit the growth of tumors during colorectal cancer therapy (Figure 5II). The tumor weight of the oxaliplatin and *F. nucleatum* treatment group was large, demonstrating the relatively poor therapeutic effect of oxaliplatin in the presence of *F. nucleatum*, which was because *F. nucleatum* could cause the chemotherapy resistance of oxaliplatin. In addition, the tumor weight and size of the Q-P-A and Q-P-A@CP[5]A and *F. nucleatum* treatment groups were relatively small, consistent with the tumor volume. Except for the oxaliplatin treatment group, there was little change in the body weight of the mice, indicating that Q-P-A@CP[5] had few side effects and a good biocompatibility during chemotherapy (Figure 5III). This supramolecular nanoparticle based on multifunctional cationic quaternary ammonium provided an effective treatment method to overcome chemotherapy-resistant cancers caused by bacteria.

Targeted and stimuli-responsive drug delivery systems show great promise in improving cancer therapy, which have the advantages of good water solubility, few side effects, and high therapeutic efficiency [137,138,139]. Hu, Zhu, Wang, and co-workers constructed supramolecular polymersomes based on water-soluble pillar[5]arene and cationic poly(glutamamide)s to deliver the hydrophilic anticancer drug mitoxantrone (MTZ) (Figure 6I) [140]. Such polymersomes with biotin ligands exhibited a good targeting ability and could specifically deliver MTZ to biotin receptor-positive cancer cells. Meanwhile, the loaded MTZ was released through the acidic environment-induced decomposition of polymersomes. In vitro experiments demonstrated that MTZ-loaded targeted polymersomes could effectively kill cancer cells and reduce their cytotoxicity to normal cells (Figure 6II,III). With outstanding therapeutic effects, the host–guest supramolecular delivery system with targeted ligands offered new opportunities for cancer therapy.

## 4. Pillararene-Based Crosslinked Supramolecular Polymers for Cancer Therapy

Crosslinked polymers usually have networks or framework structures, which can be prepared by mixing multifunctional monomers [141,142,143]. As a special class of supramolecular polymers, crosslinked supramolecular polymers combine the advantages of supramolecular polymers and crosslinked polymers [144]. Due to their excellent compatibility, mechanical properties, thermal stability, wear resistance, and creep resistance, crosslinked polymers have a wide range of practical and potential applications in adsorption and separation, catalysis, and drug delivery [145,146]. Pillararenes are recognized as the fifth generation of macrocyclic host molecules, and their rigid structure and easy functionalization endow them with unique advantages in the construction of supramolecular polymers [147]. Pillararene-based crosslinked supramolecular polymers have been increasingly investigated in the biomedical field due to their unusual architectures, fascinating properties, and interesting stimuli-responsiveness.

Template preparation is an effective “top-down” approach to obtain stable materials with the desired shape and size [148,149]. Ma and co-workers developed a crosslinked pillar[6]arene nanosponge using template preparation techniques to overcome multidrug resistance (MDR) (Figure 7I) [150]. By crosslinking pillar[6]arene and removing guest molecules, nanoparticles of uniform size could be obtained to encapsulate drugs through host–guest interactions. Because the resulting nanoparticles were made up of pillar[6]arene with unoccupied cavities, these nanoparticles were also called “nanosponges (NS)”. NS showed little cytotoxicity after incubating with HeLa cells for 48 h, indicating that NS were nontoxic to HeLa cells and NS had good biocompatibility (Figure 7II). The IC_50_ value of DOX@NS was 3.4 μM, which was significantly lower than that of free DOX (34.4 μM), indicating that NS could overcome the MDR of cancer cells (MCF-7/ADR) (Figure 7III). Mechanism studies demonstrated that the efficient loading and stable encapsulation of anticancer drugs based on host–guest interactions were the reasons for overcoming MDR. This work illustrated that the encapsulation of anticancer drugs utilizing host–guest interactions was a promising approach to overcome MDR.

Hydrogels are three-dimensional networks of crosslinked hydrophilic polymers. Supramolecular gels are an important subclass of supramolecular polymers [151,152,153,154,155]. Due to the dynamic nature of noncovalent interactions, supramolecular gels exhibit many unique and interesting properties compared with conventional covalent polymer gels, such as responsiveness to external stimuli, reversible shear sensitivity, and excellent self-healing capabilities [156,157,158]. Lin, Wang, and co-workers reported a smart hydrogel based on water-soluble pillar[6]arene as an anticancer drug carrier that exhibited a remarkable loading capacity for DOX·HCl [159]. When the polymer network G1c was soaked in the aqueous solution of pillar[6]arene, a significantly swollen hydrogel was obtained due to the formation of inclusion complexes between the pillar[6]arene and ferrocene groups. The resultant drug-loaded hydrogel with a dramatic swelling–shrinking transition exhibited a pH-triggered drug release property. Zhang, Huang, and co-workers constructed supramolecular polymer network gels using metal coordination interactions and host–guest interactions between four appendant pillar[5]arene and neutral divalent guest molecules for controlled drug release [160]. DOSY experiments revealed the formation of supramolecular polymer structures with a high degree of polymerization. Supramolecular gels exhibited a temperature and pH responsivenesses, which could be used for the controlled release of different cargoes. 

Liu and co-workers exploited a pillar[5]arene-based single-molecule-layer polymer nanocapsule for drug delivery (Figure 8I) [161]. By modifying the surface of nanocapsules with targeting peptide ligands using the host–guest interaction, a targeted smart vehicle for efficient drug delivery was obtained. MTT experiments showed that the relative survival rate of HepG2 cells was over 75%, indicating that the targeted vehicle had low cytotoxicity and good biocompatibility (Figure 8II). In addition, the smart vehicle loaded with anticancer drugs could target and penetrate into tumor cells, effectively releasing DOX to kill tumor cells, which had a good inhibitory effect on tumor cell proliferation. Compared with free DOX and DOX-nanocapsules, the absorption rate of the RGD ligand DOX-nanocap was significantly improved, indicating the efficient targeting effect of the RGD ligand modified vehicle (Figure 8III). This novel vehicle provided a good platform for cancer therapy.

## 5. Pillararene-Based Supramolecular Block Copolymers for Cancer Therapy

In addition to linear supramolecular polymers, branched supramolecular polymers, and crosslinked supramolecular polymers, supramolecular block copolymers as a special class of polymers have attracted the widespread interest of researchers. Supramolecular block copolymers, composed of two or more chemically distinct polymeric blocks linked by noncovalent bonds, are key components in a variety of applications [162,163]. By combining the structural and functional advantages of supramolecular polymers and block copolymers, such as low cytotoxicity, excellent biodegradability, and sensitive environmental responsiveness, supramolecular block copolymers have been widely used in biomedical engineering and other fields [164,165]. As an emerging class of functional materials, pillararene-based supramolecular block copolymers have demonstrated significant application prospects in cancer therapy [166].

Pillararene-based supramolecular block copolymers can be used to construct stimuli-responsive drug delivery systems, which are expected to realize the targeted drug release and selective killing of cancer cells [167,168,169,170,171]. Huang and co-workers constructed a pillararene-based amphiphilic supramolecular diblock polymer for targeted drug delivery (Figure 9I) [172]. Based on the host–guest interaction of pillararenes, amphiphilic supramolecular diblock copolymers were formed from the modified pillar[5]arene and a viologen salt. Moreover, the copolymers further self-assembled into polymersomes in water. The obtained polymersomes exhibited an excellent sensitivity to redox reactions, which could be used as an effective switch to trigger the efficient release of doxorubicin hydrochloride (DOX) from the polymersomes. Compared with the group treated with the intravenous administration of free DOX and phosphate buffer solution (PBS), the DOX-loaded polymersomes were more effective in inhibiting tumor growth (Figure 9II). In vivo experiments indicated that the therapeutic effect was preserved after the encapsulating of DOX, while the damage to normal cells was reduced (Figure 9III). This work provided a new method for constructing stimuli-responsive pillararene-based supramolecular block copolymers, which had great application potential in the field of targeted drug delivery.

Xu, Cao, Zhang, and co-workers constructed a pH- and temperature-responsive supramolecular diblock copolymer for drug delivery, which was obtained by the host–guest recognition of pillar[5]arene and viologen salts (Figure 10I) [173]. The supramolecular diblock copolymer could self-assemble into supramolecular nanoparticles, which were used for the encapsulation of PSs (pyropheophorbide-a, PhA) for PDT. When the pH value of the supramolecular nanoparticle solution was adjusted to 5, the cumulative release of PhA was 43% at 37 °C. At 25 °C, a burst release of PhA was observed within the first 2 h, and the cumulative release amount reached 82% within 11 h. This result suggested that the dual-responsive nanoparticles could effectively release PhA under pH and thermal stimulation (Figure 10II). Meanwhile, PhA-loaded nanoparticles showed a low dark toxicity to A549 cells (Figure 10III). This work paved the way for the application of a supramolecular diblock copolymer in PDT. Moreover, Li and co-workers designed and synthesized a pillar[5]arene-based nonionic supramolecular pseudoblock copolymer, which was assembled with *β*-cyclodextrin end-capped poly(acrylic acid) through host–guest interactions [174]. This supramolecular pseudoblock polymer could be used as a new platform for drug delivery and cancer therapy.

## 6. Conclusions

In summary, many remarkable pillararene-based supramolecular polymers have been developed for cancer therapy, ranging from linear supramolecular polymers, branched supramolecular polymers, to crosslinked supramolecular polymers and supramolecular block copolymers. Compared with the traditional polymer chemistry strategy of using stable and rigid covalent bonds, the construction of supramolecular polymers shows reversibility and dynamics. These properties provide the possibility to realize its stimulus responsiveness and endows supramolecular polymers with incomparable advantages such as reversibility, adaptiveness, and self-healing. In addition, the polymerization degree of supramolecular polymers can be adjusted by changing the structure, orientation, rigidity, and flexibility of monomers, thus realizing the controllable preparation of supramolecular polymers. These characteristics can effectively solve the problems faced by covalent polymers in the fields of bioimaging and drug carriers. However, many challenges remain to be addressed before these supramolecular polymers can be used for clinical applications.

(i) More efforts should be devoted to design new building blocks or develop new methods to expand the structure and type of pillararene-based supramolecular polymers. Especially, the combinatorial use of multiple noncovalent interactions deserves more attention.

(ii) Long-term monitoring of the organism is also essential. Based on the full use of imaging technology, diagnostic/imaging functions are integrated into supramolecular drugs to achieve precision therapy.

(iii) Developing novel targeted and stimuli-responsive drug delivery systems holds extraordinary potential in cancer therapy. In this way, some problems of traditional chemotherapeutic drugs can be overcome, such as poor water solubility, low therapeutic efficiency, and adverse effects on normal cells.

Although pillararene-based supramolecular polymers for clinical applications are still a long-standing challenge in biomedical development, scientists have made significant progress. It can be expected that pillararene-based supramolecular polymers will have a broad development prospect in the near future.

## Figures and Tables

**Figure 1 molecules-28-01470-f001:**
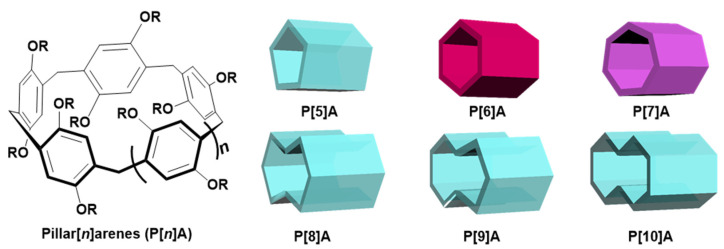
General molecular structures and schematic illustration of pillar[*n*]arenes.

**Figure 2 molecules-28-01470-f002:**
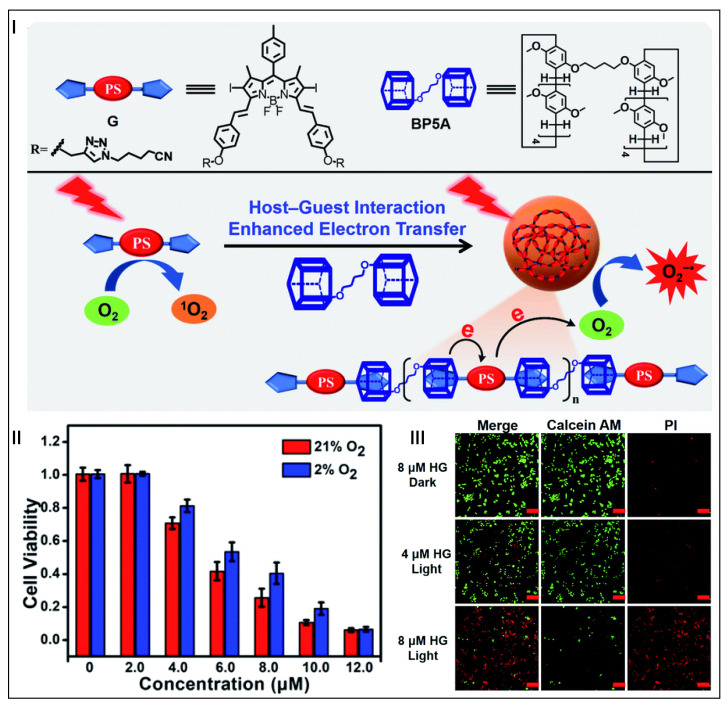
Schematic representation of the structure of BP5A and iodide BODIPY, and the schematic illustration of photo-induced generation of reactive oxygen species (**I**). Cell viability of HeLa cells at different concentrations of supramolecular polymer under normoxia or hypoxia conditions (**II**). Confocal laser scanning microscopy images of calcein AM/PI-stained HeLa cells (**III**). (reproduced with permission of Royal Society of Chemistry from ref. [106]).

**Figure 3 molecules-28-01470-f003:**
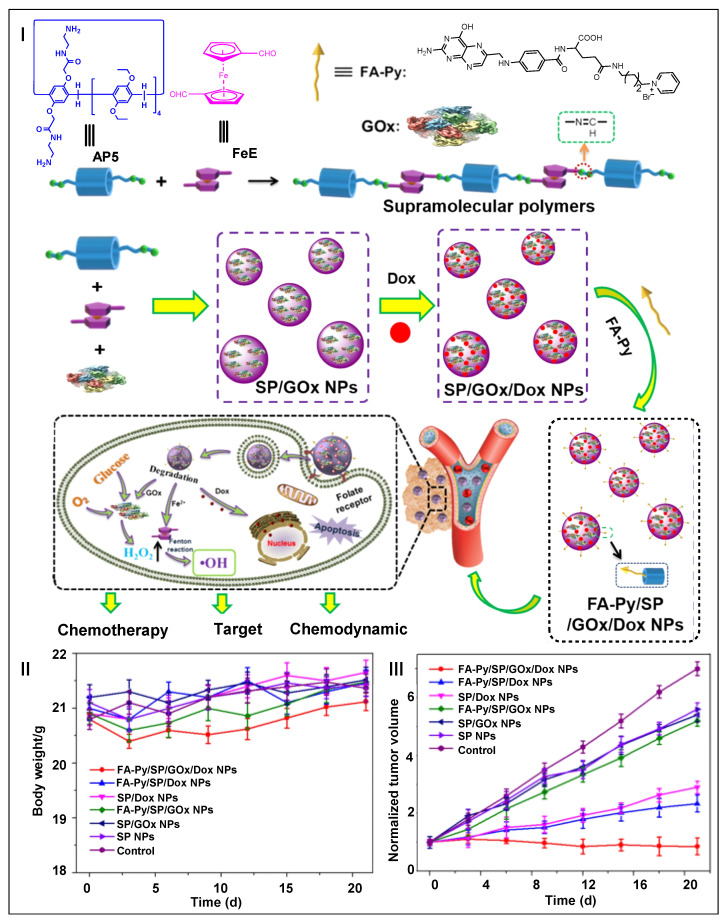
Schematic illustration of FA-Py/SP/GOx/Dox NPs in targeted synergistic chemo–chemodynamic therapy (**I**). Body weight curves of tumor-bearing mice after different formulations (**II**). Tumor growth inhibition curves of tumor-bearing mice after different formulations (**III**). (reproduced with permission of Springer Nature from ref. [111]).

**Figure 5 molecules-28-01470-f005:**
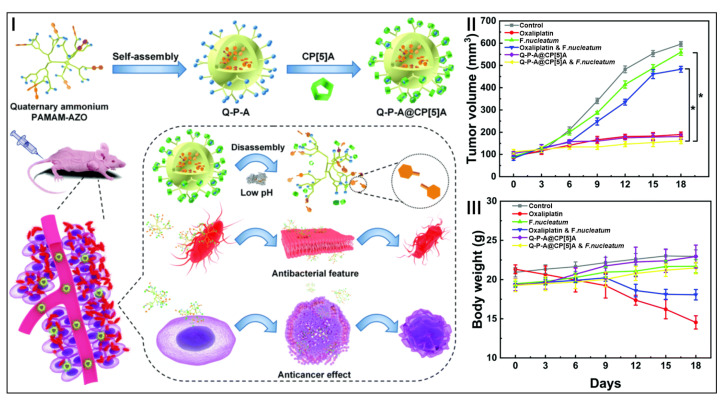
Schematic illustration of Q-P–A@CP[5]A in antibacterial and antitumor therapies (**I**). Tumor growth inhibition curves of HT29 tumor-bearing nude mice after various formulations (* *p* < 0.05) (**II**). Body weights of HT29 tumor-bearing nude mice after various formulations (**III**). (reproduced with permission of Royal Society of Chemistry from ref. [136]).

**Figure 6 molecules-28-01470-f006:**
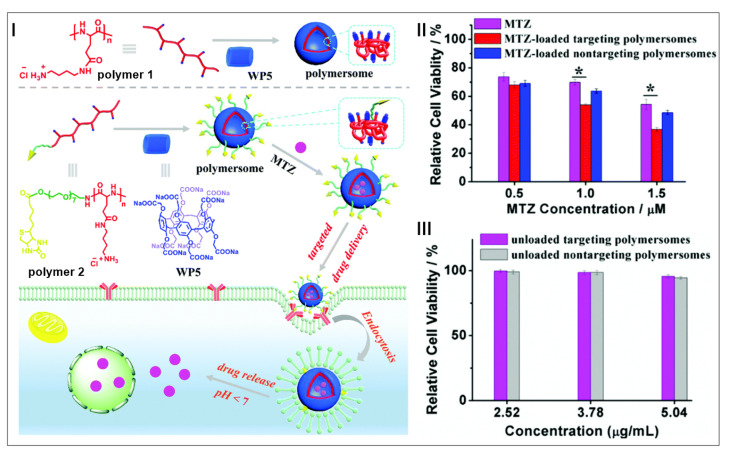
Schematic illustration of supramolecular polymersomes in targeted drug delivery (**I**). In vitro cytotoxicities of free MTZ, MTZ-loaded nontargeting and targeting polymersomes against HeLa cells (**II**). In vitro cytotoxicities of unloaded nontargeting and targeting polymersomes against HeLa cells (* *p* < 0.05) (**III**). (reproduced with permission of American Chemical Society from ref. [140]).

**Figure 7 molecules-28-01470-f007:**
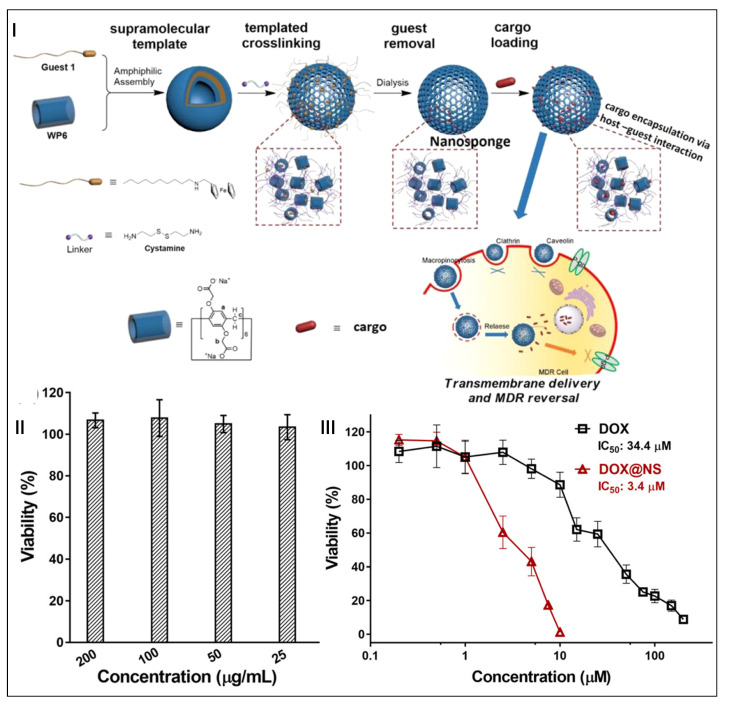
Schematic illustration of crosslinked pillar[6]arene nanosponges (NS) in overcoming MDR (**I**). Viability of HeLa cells treated with NS at different concentrations (**II**). IC_50_ plot curve of DOX and DOX-loaded NS against MCF-7/ADR cell line (**III**). (reproduced with permission of Royal Society of Chemistry from ref. [150]).

**Figure 8 molecules-28-01470-f008:**
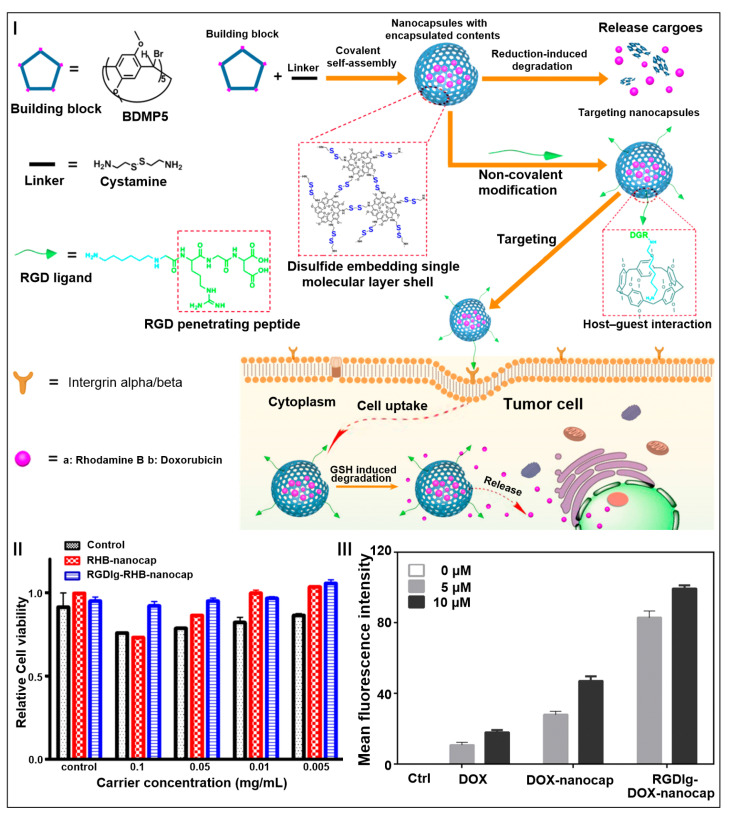
Schematic illustration of pillar[5]arene-based single-molecular-layer polymer nanocapsules in targeting anticancer drug delivery (**I**). In vitro cytotoxicity of different vehicles against HepG2 cell (**II**). Quantitative uptake of DOX, DOX-nanocap, and RGD ligand-DOX-nanocap by zebrafish larval at different concentrations (**III**). (reproduced with permission of American Chemical Society from ref. [161]).

**Figure 9 molecules-28-01470-f009:**
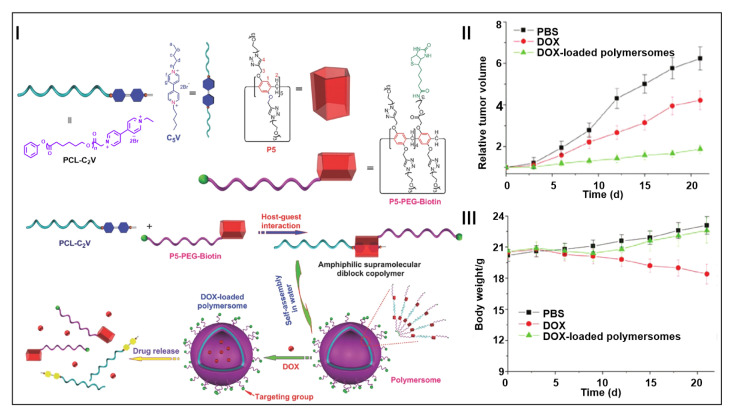
Schematic illustration of supramolecular polymersome in drug delivery (**I**). Tumor growth inhibition curves of HeLa tumor with different formulations (**II**). Body weight changes in mice bearing HeLa tumors after different treatments (**III**). (reproduced with permission of John Wiley and Sons from ref. [172]).

**Figure 10 molecules-28-01470-f010:**
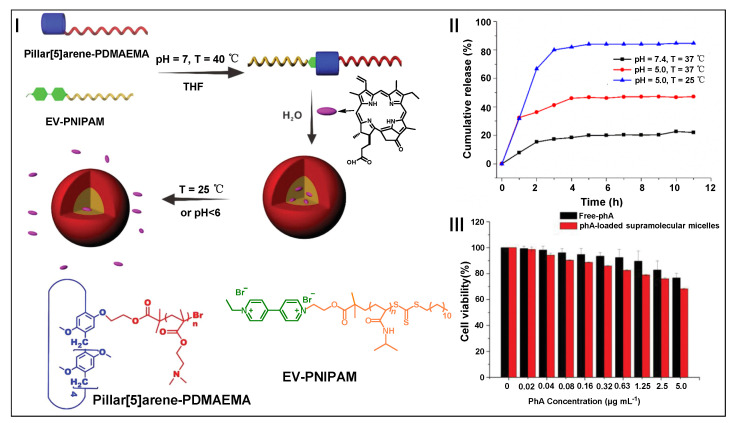
Schematic illustration of dual-stimuli-responsive supramolecular nanoparticles from a pillar[5]arene-based supramolecular diblock copolymer for PDT (**I**). Drug release profiles of supramolecular nanocarriers (**II**). The dark cytotoxicity of free PhA and PhA-loaded nanocarriers against A549 cells (**III**). (reproduced with permission of John Wiley and Sons from ref. [173]).

## Data Availability

Not applicable.

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
