# Peer review of "Pillararene-Based Supramolecular Polymers for Cancer Therapy"

_molecules, 2023, doi:10.3390/molecules28031470_

Round 1

Reviewer 1 Report

The paper presents full review on pillararene-based supramolecular polymers for cancer therapy. The work is well written and organized, the attention is drawn to the beautiful and clear figures that are illustrations for the discussed molecules. The work refers to 149 literature items that appeared in the literature mainly in the years 2012-2022.  Authors summarize in an interesting but not overloaded way the research progress of pillararene-based supramolecular polymers for cancer therapy including linear supramolecular polymers, branched supramolecular polymers, and cross-linked supramolecular polymers. The research development on pillararene-based supramolecular polymers for cancer therapy is also discussed. In my opinion, this review can be published in its current form.

Reviewer 2 Report

In this review, Zhou and Yan summarized the state of the art of pillararene-based supramolecular polymers for cancer therapy, and illustrated the developmental trend and future perspective. The examples described therein are conceivable and interesting; however, some collected information are not very substantial and even need to be corrected. Therefore, I thus recommend this work to be published in Molecules after the following revision, and there are some questions should be noted:

1. The author should carefully check the examples listed in this review. According to the concept proposed by E. W. Meijer, the supramolecular polymers refer to the ones based on monomeric units held together by directional and reversible secondary interactions (Chem. Rev., 2001, 101, 4071; Chem. Rev., 2009, 109, 5687). Therefore, the example listed in Figure 2 seemed not very reasonable.

2. Some delivery systems were not utilized as anti-tumor drug delivery systems, such as the examples listed as Figures 4a and 8a, and the authors should carefully check them.

3. The system listed in Figure 9 was not a cross-linked system, and the authors should recalibrate it carefully.

4. The molecular structure of DOX in Figure 8a was incomplete and wrong, and the size and style of the structures should be unified.

5. It is believed that the concept concerning the supramolecular polymers should be introduced in the first section. Moreover, the significance to review this topic should also be highlighted and strengthened in this section.

Reviewer 3 Report

The manuscript reviews a very interesting topic centered on the use of supramolecular systems in cancer therapy. The systems presented in this review are based on pillarene polymers and the authors considered the linear supramolecular polymers, together with branched and cross-linked supramolecular polymers. The topic is actual, and the potential applications of these systems deserve this deepening.

However, the paper can not be published in the present form.

A main comment that needs to be considered from the authors is that nine Figures out of ten are taken from other manuscript. I think that the authors, in preparing the manuscript, should not limit their work to reproduce figures taken from other papers, but they should create themselves figures that can help the reader to understand the concepts expressed in the text. Some occasional Figure can be inserted from published material, but this cannot be the rule.

Together with this main comment, some minor point needs to be addressed:

Page 2, lines 69-75. The two phrases are not linked together. “And copolymers…” can not be the starting of a phrase.

Page 3, Figure 2. The graphic quality of panel II and III in the figure 2 is low. It needs to be enhanced.
